# *Toxoplasma gondii* seropositivity among patients with sickle cell disease: Prevalence and association with blood transfusion history

Verner N. Orish[1]*, Renosten E. Tetteh[2], David Adzah[1],
Chinecherem A. Ndiokwelu[3], Emmanuel A. Allotey[3], Evans A. Yeboah[2],
Sylvester Y. Lokpo[3], Kenneth Ablordey[3], Duneeh R. Vikpebah[3], Ekene K. Nwaefuna[4],
Precious K. Kwadzokpui[5], Noble D. Dika[6], Elom Y. Dzefi[7], Kokou H. Amegan-Aho[8],
Aninagyei Enoch[9], Senyo Tagboto[10]

1 Department of Microbiology and Immunology, School of Medicine, University of Health and Allied Sciences, Ho, Volta Region, Ghana, 2 Department of Virology and Immunodiagnostics, Ho Teaching Hospital-Laboratory, Ho, Ghana, 3 Department of Medical Laboratory Sciences, School of Allied Health Sciences, University of Health and Allied Sciences, Ho, Ghana, 4 Radiation Entomology and Pest Management Center, Biotechnology and Nuclear Agriculture Research Institute, Ghana Atomic Energy Commission, Accra, Ghana, 5 Laboratory Department, Ho Teaching Hospital, Ho, Volta Region, Ghana, 6 Department of Hematology, Ho Teaching Hospital-Laboratory, Ho, Ghana, 7 Department of Blood bank Services, Ho Teaching Hospital-Laboratory, Ho, Ghana, 8 Department of Paediatrics, University of Health and Allied Sciences, Ho, Ghana, 9 Department Biomedical Sciences, School of Basic and Biomedical Sciences, University of Health and Allied Sciences, Ho, Ghana, 10 Department of Internal Medicine, University of Health and Allied Sciences, Ho, Ghana

☯ These authors contributed equally to this work.
* vorish@uhas.edu.gh

## Abstract

### Background

*Toxoplasma gondii* (*T. gondii*) is a successful protozoan parasite infecting up to a third of the human population. It has varied transmission routes including ingestion of food and water contaminated by cat feces containing oocysts of the parasite and ingestion of bradyzoites in poorly cooked meat. Blood transfusion is another possible route of transmission especially among people with medical conditions requiring blood transfusion, such as those with sickle cell disease (SCD). This study aimed at finding out the prevalence of *T. gondii* infection and the association of blood transfusion among patients with SCD.

### Method

This study was a cross-sectional study involving SCD patients attending the SCD clinic at the Ho Teaching Hospital in the Volta Region of Ghana. Questionnaire administration was employed to obtain sociodemographic information, cat ownership, consumption of poorly cooked meat, as well as blood transfusion history. A blood sample was collected and anti-*T. gondii* IgG and IgM were detected using Rapid Diagnostic Test (RDT), while Enzyme-linked Immunosorbent Assay (ELISA) was

**Data availability statement:** All relevant data are within the paper and its Supporting Information files.

**Funding:** The author(s) received no specific funding for this work.

**Competing interests:** The authors have declared that no competing interests exist.

used as the gold standard and reference. Seropositivity was defined as either positive for IgG, IgM or both. Data was analyzed using SPSS version 23, with frequency distribution done for the sociodemographic variables and the prevalence of RDT and ELISA anti-*T. gondii* IgG and IgM. Pearson Chi-square analysis was performed to find any significant association between diagnosis of *T. gondii* infection with sociodemographic variables and blood transfusion. Logistic regression analysis was performed to investigate the odds of seropositivity (ELISA) with sociodemographic variables and blood transfusion.

## Results

A total of 156 SCD patients participated in this study of which 124 (79.5%) and 32(20.5%) were HbSS and HbSC respectively. Among the study participants, 105 (67.3%) had a history of blood transfusion. A total of 60 (38.5%) and 83 (53.2%) patients were positive for RDT and ELISA respectively. No significant association was seen between *T. gondii* diagnosis and cat ownership (RDT,20[37.7%], p = 0.891; ELISA, 27[50.9%], p = 0.673) and consumption of poorly cooked meat (RDT,37[41.6%],p = 0.370;ELISA,53[59.6%], p = 0.211). However there was a significant association between *T. gondii* diagnosis and age, with seropositive results predominantly seen among older patients (≥20 years) (RDT, 38[52.1%], p = 0.002; ELISA 49 [67.1%, p = 0.002]. Blood transfusion had a significant association with *T.gondii* diagnosis (RDT, p = 0.003; ELISA, p = 0.001). A total of 66 (62.9%) of SCD patients who had history of blood transfusion tested positive for ELISA and they had 3 times the odds of testing positive for ELISA (adjusted OR 3.14[95% CI 1.50–6.58]; p = 0.002).

## Conclusion

The prevalence of *T. gondii* infection was higher by ELISA (53.0%) than by rapid diagnostic testing (RDT) (38.5%), and sickle cell disease patients with a transfusion history had higher odds of seropositivity. These findings highlight the need to strengthen transfusion safety protocols and consider screening strategies for *T. gondii* among high-risk populations such as patients with sickle cell disease. Also, there is the need for longitudinal research to help elucidate the true contribution of blood transfusion transmission of *T. gondii* since a cross-sectional study, causality could not be established.

---

## Introduction

*Toxoplasma gondii* (*T. gondii*) is a wide spread protozoan parasite infecting human and a wide range of warm-blooded animals [1]. Globally, it is estimated that 30–50% of the population has been exposed, with approximately 190,000 congenital toxoplasmosis occurring annually worldwide [2]. The burden varies by population: prevalence among pregnant women is around 1% [3,4], ocular toxoplasmosis affects about 2%

in the United States [5], and cerebral toxoplasmosis remains a major opportunistic infection in HIV/AIDS patients in low-income countries [6]. Transmission occurs through ingestion of food and water contaminated with oocysts from cat feces or bradyzoites in undercooked meat, less common routes such as organ transplantation and blood transfusion have also been documented [6,7].

Blood transfusion is a lifesaving intervention used to correct anemia in both adult and pediatric patients [8]. Despite routine screening, transfusion-transmitted infections (TTIs) remain a significant concern [9]. Although *T. gondii* can be transmitted via transfusion, it is not routinely screened in most resource-limited settings, making transfusions a potential source of infection [10–12]. Patients who frequently receive transfusions such as those with hematological disorders including sickle cell disease are particularly vulnerable to TTIs, including pathogens not routinely screened like *T. gondii* [13,14].

Sickle cell disease (SCD) is one of the most common hematological disorders in sub–Saharan Africa, affecting a significant proportion of the population [15]. Its clinical manifestations are often severe, leading to frequent hospital visits, admission and therapeutic interventions, including blood transfusion [16]. Patients with SCD are among the most frequent users of transfusion services in many health facilities across the region, and cases of TTIs have been reported [17].

In Ghana, several studies have suggested that *T. gondii* might be hyperendemic in the population, with several of these reporting a prevalence of above 70% in the adult population, especially among pregnant women [18]. SCD is also a very common hemoglobinopathy in Ghana, with a significant proportion of the population with the condition and about 2% of all newborns having the disease [19]. Many of these patients require several hospital visits for the management of acute episodes or crises, which often involve blood transfusions. No previous study in Ghana has looked at *T. gondii* infection in SCD or have reported the prevalence of *T. gondii* infection in the general population of the Volta Region of Ghana. This study therefore aimed to determine the prevalence of anti-*Toxoplasma* gondii antibodies among SCD patients in the Volta Region of Ghana and to assess the association between seropositivity and history of blood transfusion.

## Methodology

### Study design

This was a cross-sectional descriptive study involving SCD patients attending the SCD clinic in Ho Teaching Hospital of the Volta Region of Ghana.

### Study Area

This study was conducted at the SCD clinic of the Ho Teaching Hospital of the Volta Region. The Volta Region is one of the 16 regions in the country. The Region is located between latitudes 50 45"N and 80 45"N along the southern half of the eastern border of Ghana, which it shares with the Republic of Togo. The region shares boundaries to the west with the Greater Accra, and Eastern regions, to the north with Oti region, and has the Gulf of Guinea to the south. The Ho Teaching Hospital is the main referral facility in the Volta Region with over 300 beds capacity and comprises 5 major departments including internal medicine, surgery, obstetrics and gynecology, Child health, and Public health. It serves patients from Ghana and the neighboring West African Countries. The SCD clinic has about 210 registered SCD patients including children and adults.

### Study population

This study was conducted among the SCD patients that attend the SCD clinic of the Ho Teaching Hospital. These included both children and adults from the ages of 2–65 years. Those who were in crisis at the time of visiting the clinic, were excluded from the study.

## Sample size determination

The Cochran formula was used to calculate the sample size, using P as 50% which is an estimated prevalence of *T. gondii* infection among patients with sickle cell disease (since the actual prevalence is unknown), Z value of 1.96 with a confidence interval of 95%, and allowable error of 0.05.

$$n = \frac{z^2 \times p \times (1-p)}{e^2} = 385$$

$$n = (z^2 * p * (1-p)) / e^2 = 385$$

A modified Cochran formula was then used, imputing n = 385 as the Cochran calculated sample size with population (N) of 210 registered sickle cells disease patients that attend the clinic.

$$n = \frac{n_0}{1 + \frac{n_0 - 1}{N}} = \frac{385}{1 + (\frac{385-1}{300})} = 136$$

$$= 385/1 + (384/210) = 136$$

Thus, the minimum sample size of patients with sickle cell disease in this study was 136; however, to increase the power of the study, a sample size of 156 was collected.

## Sampling and recruitment procedure

Given the relatively small clinic population of 210 registered patients, a consecutive sampling was employed, where all eligible SCD patients attending the clinic during the study period were recruited sequentially until the required sample size was achieved. This approach was chosen over convenience sampling to minimize selection bias, as it systematically included all eligible patients rather than relying on arbitrary availability and researchers' discretion. While this method enhances representativeness within the clinic population, the generalizability of findings beyond the clinic setting remains limited.

Before recruitment all patients went through the routine clinic visit procedure which includes vital signs assessment like blood pressure, temperature, weight, respiratory rate and oxygen saturation. After this, each patient saw the attending clinician for history and physical examination. Any patient found to be clinically unstable or in crises, was excluded from the study and managed appropriately. Patients with the clinical history of HIV comorbidity were also excluded from the study. Those who were stable were spoken to and upon consent were taken through data collection involving questionnaire administration and blood sample collection. Data collection took place from 7TH August to 18TH December 2024.

## Determination of haemoglobin variant among SCD patients

Haemoglobin (Hb) electrophoresis test is routinely done on all SCD patients who enroll into the clinic, to confirm their SCD variant. The Haematology Unit of the Ho Teaching Hospital uses the cellulose acetate paper technique in alkaline pH (Hb electrophoresis). This predominantly report Hb variants such as HbS, and HbC. In brief, haemoglobin phenotypes were determined using cellulose acetate alkaline electrophoresis (pH 8.6). Haemolysate was prepared by saline-washing of RBCs and lysing with 0.5% (v/v) Triton X-100 in 100 mg/L potassium cyanide. Electrophoresis was run on cellulose acetate membrane soaked in Tris-EDTA Borate (TEB) buffer (pH 8.4–8.6) by passing direct current delivering 350V at 50mA through the electrophoresis tank for 25 minutes. A positive control containing haemoglobin C, S, F, and A was electrophoresed with each batch during phenotyping.

## Questionnaire validation and administration

The questionnaire used in the data collection was originally developed for this study as a structured closed-ended questionnaire. After the design and before data collection a detailed face and content validation was done by two independent experts who reviewed the comprehensiveness of the questions, appropriateness, relevance, and objectiveness. The questionnaire was deemed suitable and appropriate in meeting the study objectives.

Questionnaires were administered to the SCD patients to obtain sociodemographic and socioeconomic information, such as age, sex, educational and occupational status, marital status, and others. Information on the lifetime history of blood transfusion was obtained. This was defined as having ever received one or more transfusions of any type, with responses categorized as "Yes" or "No". To ascertain the overall risk of *T. gondii* infection, the following information were obtained; (1) Information on consumption of well or poorly cooked meat (pork, beef chicken, snail fish etc.), and raw vegetables (2) source of drinking water

## Blood sample collection

Five milliliters of blood was collected and distributed between an EDTA tube and a gel separator tube (appropriately labelled with participant unique ID). The EDTA whole blood was spun at 2000 rpm for 5 minutes to separate the plasma. Serum was extracted from the whole blood that was placed into cryo-tubes (appropriately labelled with participant's unique ID) by spinning at 3500 rpm for 5 minutes and subsequently stored at −20 °C until further analysis.

## Detection of *T. gondii* Antibodies Using Rapid Diagnostic Test (RDT) and ELISA Kits

*T. gondii* immunoglobulins G (IgG) and M (IgM) were detected using a 2-step approach. Firstly, a qualitative membrane strip-based immunoassay RDT from Shanghai EIDERE Medical Technology Co. Ltd, China (LOT: 2024060602; manufactured 2024-06-06; expiration 2026-06-05) and subsequently, an ELISA panel from the same manufacturer (LOT: 202406; manufactured 2024−06; expiration 2025−06). All assays were performed according to the manufacturer's instructions. Quality control measures for ELISA included the use of positive and negative controls with each batch, adherence to recommended storage conditions (2–8°C), and duplicate testing of 10% of samples to confirm consistency. Quality control for RDTs was ensured through manufacturer-provided positive and negative controls with each batch, and repeat testing was performed in cases of invalid or discordant results.

The test was based on the antigen-antibody interaction of toxoplasma-specific antibodies with precoated toxoplasma antigens. Regarding the RDT, the nitrocellulose membrane of the kit was precoated with the antigen so that if the patient`s plasma contains the toxoplasma antibody, an immune complex would be formed. This immune complex is then captured as a burgundy band of the test region on the nitrocellulose membrane. All RDT tests were validated by the presence of a control band, which detected human immunoglobulins.

## Rapid diagnostic test (RDT) procedure

Per the manufacturer's instructions, the test kit was removed from its pouch and laid flat on the workbench. A vertical dropper was used to aspirate approximately 10 µL of patient plasma and subsequently dispensed into the sample well. Two drops of the sample buffer (approximately 70 µL) were dispensed into the same sample well. A timer was then set and a determination was made after 15minutes. For positive results, there was a burgundy colour development at both control and test band regions for either IgG or IgM or both IgG & IgM. For negative results, an absence of colour band development at the test line region along with a single band development at the control region was inferred as negative. For this study, RDT positive is defined as positive for either IgM, IgG or IgM and IgG combination.

## Enzyme-linked immunosorbent assay (ELISA) procedure

Stored-frozen serum was allowed to thaw at room temperature. The plate was sectioned to include wells for both positive and negative controls, a blank, and a set of standards in each run. Each sample was run in triplicate, and the average of the absorbances were calculated. For each sample, 10 μL of the sample was mixed with 40 μL of the sample diluent and dispensed into their respective wells. To each well, 100 μL of HRP conjugate was added and incubated at 37°C for 60 minutes. After incubation, each well was washed serially and after the fifth round of wash, the supernatant was decanted and the wells were blotted with a clean dry tissue paper. About 50 μL of commercially prepared chromogen A and later B were added and mixed. This mixture was left to incubate at 15minutes at 37°C. A stop solution of 50 μL was then added to the well to halt the reaction and colour development. The resultant concentration of the mixture was measured at the manufacturer prescribed 450nm. The concentration was deduced from comparing the measured absorbance in each sample well to that of the standards which were run together with the samples. To determine the critical cut-off, the average absorbance of the negative controls was summed with 0.15. A negative result for IgG or IgM was defined as; sample absorbance < calculated critical cut off. A positive result for IgG or IgM was defined by sample absorbance ≥ calculated critical cut off. IgG avidity testing was not performed due to budgetary constraints. However, the focus was on the overall exposure or evidence of infection, rather than distinguishing between recent and past infections. ELISA positive is defined as positive for either IgM, IgG or IgM and IgG combination. For this study, seropositivity was operationally defined using ELISA results, and this served as the primary outcome of interest.

## Data analysis

Data were analyzed using IBM SPSS Statistics version 25 (IBM Corp., Armonk, NY, USA). Prior to analysis, all datasets were checked for completeness, consistency, and data cleanliness to ensure accuracy and reliability. Categorical variables such as gender, occupation, educational status, blood transfusion history, cat ownership and others were assigned numeric codes, while continuous variables (age) were retained in their original scale. Recategorization or combining categories were done for age and educational status to avoid sparse cells and low event per variable. There were no missing values across the dataset; therefore, all cases were included in the analysis without the need for imputation or deletion.

The diagnostic performance of the rapid diagnostic test (RDT) was evaluated against ELISA as the reference standard, using sensitivity, specificity, positive predictive value (PPV), negative predictive value (NPV), and diagnostic efficiency.

Descriptive statistics, including frequency distributions, were used to summarize sociodemographic characteristics, blood transfusion history, and the prevalence of *Toxoplasma gondii* infection among sickle cell disease (SCD) patients. Pearson's chi-square test was used to assess associations between *T. gondii* seropositivity and variables such as cat ownership, consumption of undercooked meat, and blood transfusion history.

Multiple logistic regression was performed to estimate the association between blood transfusion and the odds of *T. gondii* ELISA seropositivity. Unadjusted and adjusted odds ratios (ORs) with 95% confidence intervals (CIs) were estimated. Covariates were mainly selected a priori based on theoretical and clinical relevance and prior literature, including age, education level, cat ownership and eating undercooked or raw meat. Age and education were included to account for differences in exposure risk and health literacy, coincidentally they were statistically significant in chi-square analysis. Conversely, cat ownership and eating of undercooked meat were not statistically significant but were included due to their established role in *Toxoplasma gondii* transmission. No automated variable selection methods were used. To reduce the risk of overfitting considering the limited number of ELISA-positive cases, we strategically combined categories of key variables to minimize the number of model parameters. For example, age groups were collapsed into combined categories of ≤19 years, and ≥20 years while education levels were regrouped into Basic (No education, Primary, JHS), and post Basic (SHS and Tertiary). This is to ensure that the events-per-variable (EPV) is > 10 in the full model. In addition, clinically plausible interaction between age and transfusion was pre-specified and tested in the full model. When the interaction

term reached statistical significance, stratum-specific odds ratios were calculated and reported separately for participants ≤19 years and ≥20 years.

All statistical analyses were conducted at a 95% confidence interval, and p-values ≤ 0.05 were considered statistically significant.

## Ethical considerations

Ethical clearance was obtained from the University of Health and Allied Sciences Research Ethics Committee (UHAS-REC A.8 17123−24). Strict confidentiality was employed during data collection of the study. Written informed consent was obtained from all participants and assent was obtained from older children before commencement of the study. Participants were informed that participation was solely voluntary and they are free to discontinue with the study at any point without any consequences. Results of serological testing were communicated to both the attending physician and the patients. All participants received education on toxoplasmosis and its possible transmission routes. Those who tested positive were appropriately counseled, and follow-up was offered in accordance with the clinical judgment of the attending clinician.

## Results

A total of 156 SCD patients were recruited into this study. Of these majority were, teenagers from the ages 13–19 years (60, 38.5%), females (88, 56.4%), had secondary high school (SHS) (64, 41.1%) and the majority of the participants were unemployed students (103, 66.02%) (Table 1). The majority of the participants, were unmarried (135, 86.5%), lived in urban centers (81, 51.9%), residence without fence or walls (91, 58.6%) and compound not cemented (107, 68.6%) (Table 1). Almost all participants had potable water with majority on sachet water (98, 62.8%) and pipe-borne water (56, 35.9%) (Table 1). Table 1 presents the overall (marginal) distributions of sociodemographic characteristics.

Table 2 shows the clinical characteristics and blood transfusion (B.T) history among the participants. Haemoglobin SS was the most common genotype reported in 124 participants (79.5%). Majority of them have had blood transfusion (105, 67.3%), but fewer of them have had blood transfusion in the past 12 months (35, 33.3%). Severe anaemia was the major reason for blood transfusion among participants in this study (69, 65.7%). Table 2 presents the overall (marginal) distributions of clinical characteristics and transfusion history.

Fig 1 shows the percentage distribution of blood transfusion among the SCD participants. Among the various age groups of SCD patients those between the ages of 46–59 years had the highest of percentage blood transfusion (87.5%) with the least seen among those who are 60 years and above (50%). However, this finding was not significant (p = 0.28). More males (75%) had blood transfusion than females (61.4%) (p = 0.07). More participants from rural (76.7%) areas had more transfusion than those from urban areas (59.3%) (0.07). There was a significant difference in blood transfusion distribution among SCD patients (p ≤ 0.001) with more blood transfusion among HBSS (78.2%) compared to those with HBSC (36.4%).

Fig 2 highlights some other risk factors for *T. gondii* infections. Cat ownership was reported for only 53 (33.9%) participants but 92 (58.9%) participants encounter or saw stray cats in their residences. Majority of the participants, consume undercooked meat (87, 55.8%), and practice hand hygiene (154, 98.7%) with most of them washing hands frequently (139, 90.3%) while a few do it occasionally (15, 9.7%).

Fig 3 highlights the laboratory diagnosis of *T. gondii* infection with ELISA and RDT used in this study. While both RDT and ELISA did not differ much in IgG detection (RDT, 49 [31.4%]; ELISA 50,[32.1%]), ELISA reported more positives for Ig M and IgG combination (25 [16%]), compared to RDT (3,[1.9%]. Both RDT and ELISA reported 8(5.1%) positive for IgM alone. (Fig 3).. Overall, RDT reported fewer positive (60, 38.5%) compared to ELISA which reported 83 positives(53.2%) of the 156 total SCD patients.

**Table 1. Sociodemographic characteristics among SCD patients.**

| Variables | Frequency | Percentage (%) |
|---|---|---|
| Total | 156 | 100.0 |
| **Age** | | |
| ≤5 | 10 | 6.4 |
| 6-12 | 13 | 8.3 |
| 13-19 | 60 | 38.5 |
| 20-26 | 40 | 25.6 |
| 27-44 | 23 | 14.7 |
| 45-59 | 8 | 5.1 |
| ≥60 | 2 | 1.3 |
| **Gender** | | |
| Male | 68 | 43.6 |
| Female | 88 | 56.4 |
| **Education** | | |
| No Education | 4 | 2.5 |
| Primary | 20 | 12.8 |
| JHS | 27 | 17.3 |
| SHS | 64 | 41.1 |
| Tertiary | 41 | 26.3 |
| **Occupation** | | |
| Unemployed | 17 | 10.9 |
| Student | 103 | 66.02 |
| Trader | 12 | 7.7 |
| Farmer | 1 | 0.6 |
| Civil servant | 19 | 12.2 |
| Artisan | 4 | 2.7 |
| **Marital status** | | |
| Single | 135 | 86.5 |
| Married | 19 | 12.2 |
| Widowed | 2 | 1.3 |
| **Residence** | | |
| Urban | 81 | 51.9 |
| Rural | 75 | 48.1 |
| **Water supply** | | |
| Sachet water | 98 | 62.8 |
| Pipe-borne | 56 | 35.9 |
| Borehole | 1 | 0.6 |
| Rainwater | 1 | 0.6 |
| **Walled residence** | | |
| Yes | 65 | 41.7 |
| No | 91 | 58.3 |
| **Cemented compound** | | |
| Yes | 49 | 31.4 |
| No | 107 | 68.6 |

Frequencies (n) and percentages (%) of age, gender, education, occupation, marital status, residence, water supply, and housing characteristics among 156 patients with SCD. Overall marginal distribution is shown, stratified conditional distributions by *T. gondii* RDT and ELISA seropositivity are presented in Table 4.

**Table 2. Clinical Characteristics and Blood Transfusion History.**

| Variables | Frequency | % |
|---|---|---|
| **Genotype (n = 156)** | | |
| HbSS | 124 | 79.5 |
| HbSC | 32 | 20.5 |
| **Hydroxyurea Intake (n = 156)** | | |
| Yes | 85 | 54.5 |
| No | 71 | 45.5 |
| **Blood transfusion (n = 156)** | | |
| Yes | 105 | 67.3 |
| No | 51 | 32.7 |
| **Reason for blood transfusion (n = 105)** | | |
| Acute pain | 10 | 9.5 |
| Severe anaemia | 69 | 65.7 |
| Acute pain+ severe Anaemia | 14 | 13.3 |
| Other complication | 6 | 5.7 |
| Unknown | 6 | 5.7 |

Genotype distribution, hydroxyurea intake, transfusion history, and reasons for transfusion among 156 patients. Overall marginal distribution is shown. Stratified conditional distributions by *T. gondii* RDT and ELISA seropositivity are presented in Table 5.

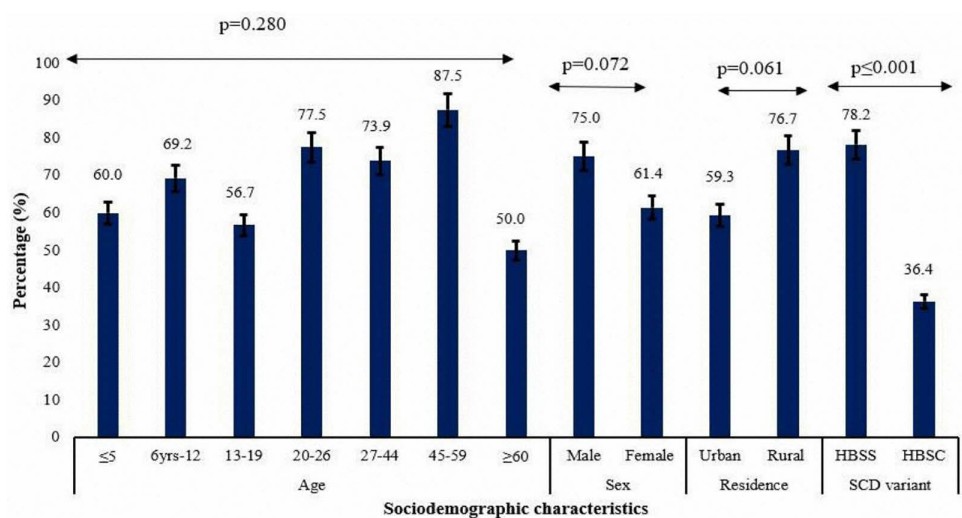

**Fig 1. % Distribution of Blood transfusions among SCD patients.**

Table 3 highlights the diagnostic performance of RDT compared with the ELISA as the reference standard. Compared to the ELISA technique, the sensitivity of the RDT was 68.67% (95% CI: 57.56–78.41%) and a specificity of 95.89% (95% CI: 88.46–99.14%). The positive predictive value (PPV) was 95.00% (95% CI: 86.14–98.31%), and the negative predictive value (NPV) was 72.92% (95% CI: 66.11–78.79%). Overall efficiency was 81.41% (95% CI: 74.41–87.18%). Agreement between the RDT and the ELISA (the reference standard), as measured by Cohen's kappa, was 0.634 (95% CI: 0.513–0.754), indicating substantial agreement beyond chance, according to Landis & Koch scale. The confidence intervals

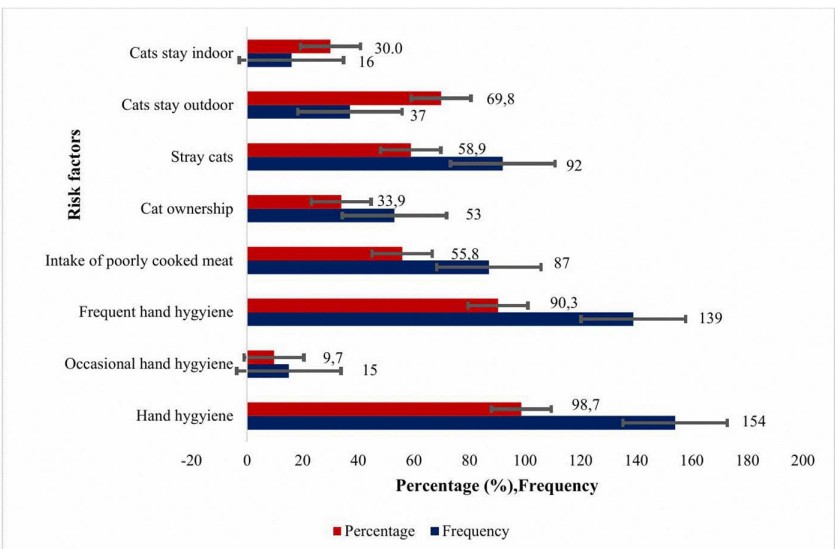

**Fig 2. Other Risks factor for *Toxoplasma gondii* infection.**

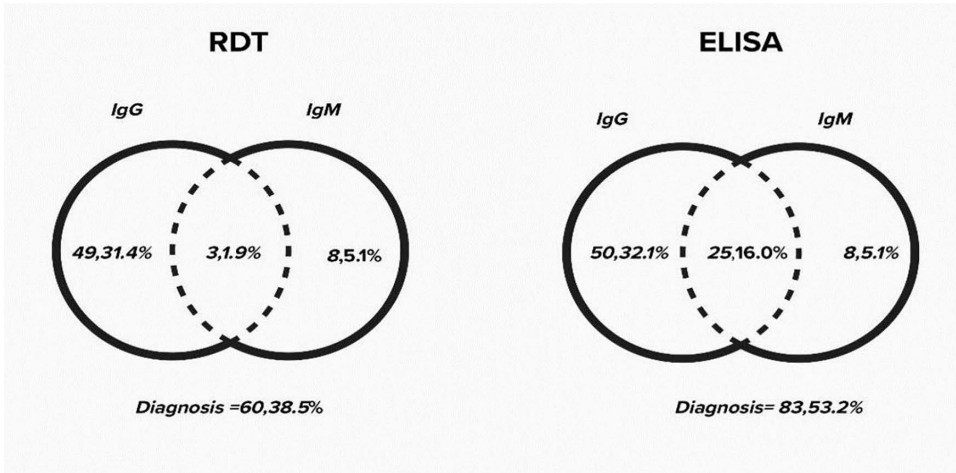

**Fig 3. Laboratory diagnosis of Toxoplasma gondii infection with RDT and ELISA.**

were computed using the exact binomial (Clopper–Pearson) method, which provides conservative bounds appropriate for proportions derived from binary outcomes.

Table 4 stratifies the socio-demographic characteristics of the patients by RDT and ELISA diagnosis for *T. gondii*. Statistical comparisons were performed using pearson chi-square test, and prevalence ratios (PR) with 95% confidence intervals (CI) were calculated to estimate associations. A significant association was observed between participants' age and the results of ELISA (p = 0.002) and RDT (p < 0.002). This table presents conditional distributions (positive vs negative results) alongside overall totals (marginal distributions). Combining the various age categories into ≤19 for younger age groups and ≥20 years for older participants, older participants had 97% and 64% higher prevalence of RDT and ELISA seropositivity respectivvely. (RDT PR 1.97 [95% CI, 1.27–3.06]), (ELISA PR 1.64[95% CI, 1.13–2.38]). Combining

**Table 3. Diagnostic efficiency of RDT compared with ELISA for detecting *T. gondii* antibodies.**

| Laboratory diagnosis | RDT +VE | RDT –VE | TOTAL |
|---|---|---|---|
| ELISA +VE | 57/TP | 26/FN | 83 |
| ELISA – VE | 3/FP | 70/TN | 73 |
| TOTAL | 60 | 96 | 156 |
| Index | Estimates | 95% CI (Clopper–Pearson | |
| Sensitivity | 68.67 | 57.56 - 78.41 | |
| Specificity | 95.89 | 88.46 - 99.14 | |
| NPV | 72.92 | 66.11 - 78.79 | |
| PPV | 95.00 | 86.14 - 98.31 | |
| Efficiency | 81.41 | 74.41 - 87.18 | |
| Kappa value | 0.634 | 0.513–0.754 | |

Cross-tabulation of test results with sensitivity, specificity, predictive values, efficiency, and kappa statistic.

TP = True positive, FN = False negative, FP = false positive, TN = True negative.

the various groups in education into Basic (No education, Primary and JHS) and post Basic (SHS and Tertiary), participants with post Basic education had about 53% higher prevalence of ELISA seropositivity (ELISA, PR 1.53[95% CI, 1.08–2.16])

Table 5 stratified the clinical characteristics and blood transmission history of the patients by RDT and ELISA diagnosis for *T. gondii*. Statistical comparisons were performed using pearson chi-square test, and prevalence ratios (PR) with 95% confidence intervals (CI) were calculated to estimate associations. This table presents conditional distributions (positive vs negative results) alongside overall totals (marginal distributions).Significance was seen with blood transfuion history with both RDT (p = 0.003) and ELISA (p = 0.001). More partipants who had blood transfusion tested positive for ELISA (66, 62.9%), and those who did not have blood transfusion significantly had a 47% lower prevalence of seropositivity with ELISA (PR 0.53 [95% CI 0.35–0.81]. Significantly more SCD patients who had blood transfusion because of chronic anaemia tested positive for ELISA (42, 60.9). However, those with reasons of acute anaemia showed a significant 31% higher prevalence of seropositivity with ELISA [8, 80%] (PR 1.31[95% CI 1.01–1.70].

Multiple logistic regression analysis was further carried out on variables with significant association with ELISA resuts to find out the odds of positive ELISA results among the SCD ptients. Table 6 showed that patients who had B.T significantly had 3 times the odds of testing positive with ELISA (unadjusted OR 3.39[95% CI 1.67–6.68]; p = 0.001)and this odd remained after adjusting for age, level of educational level, cat ownership and eating undercooked meat (adjusted OR 3.39[95% CI 1.60–7.19]; p = 0.001). (Table 6).

### Diagnostic Performance and Validation Metrics of a Toxoplasmosis Seropositivity Prediction Model

The adjusted logistic regression model predicting *Toxoplasma gondii* seropositivity demonstrated moderate discriminative ability, AUC = .72, 95% CI [.64,.80]. Calibration was acceptable, reflected by a Brier score of.21 and a Brier skill score of.16, indicating improvement over baseline predictions. The Hosmer–Lemeshow goodness-of-fit test was non-significant, $\chi^2(8) = 7.62$, p = .42, suggesting adequate model fit. Explained variance was moderate (McFadden's $R^2$ = .12; Nagelkerke $R^2$ = .21). Multicollinearity was not a concern, with a mean variance inflation factor (VIF) of 1.17 across predictors. Overall, the model showed balanced performance across discrimination, calibration, and fit metrics, supporting its validity for predicting toxoplasmosis seropositivity in this sample (N = 156).

**Table 4. Socio-demographic characteristics of SCD patients stratified by laboratory diagnsis of *T. gondii* infection by RDT and ELISA.**

| Characteristics | RDT | | | | ELISA | | | | |
| | Positive (%) | Negative (%) | P-value | PR (95% CI) | Positive (%) | Negative (%) | Total, n [%] | p-value | PR (95% CI) |
|---|---|---|---|---|---|---|---|---|---|
| **Age yrs** | | | | | | | | | |
| **Age yrs** | | | | | | | | | |
| +≤19 | 22[26.5] | 61[73.5] | 0.002 | 1.97(1.27-3.06) | 34 [41.0] | 49 [59.0] | 83[53.2] | 0.002 | 1.64(1.13-2.38) |
| ≥20 | 38[52.1] | 35[47.9] | | | 49 (67.1) | 24 (32.9) | 73[46.8] | | |
| **Gender** | | | | 1.16(0.87-1.54) | | | | | 1.06(0.83-1.35) |
| Male | 24[35.3] | 44[64.7 | 0.482 | | 35[51.5] | 33[48.5] | 68[43.6] | 0.701 | |
| Female | 36[40.9] | 52[59.1] | | | 48[54.5] | 40[45.5] | 88[56.4] | | |
| **Education** | | | | | | | | | |
| Basic | 16[31.4] | 35[68.6] | 0.312 | 1.34(0.84-2.13) | 20[39.2] | 31[60.8] | 51[32.7] | 0.033 | 1.53(1.08-2.16) |
| Post basic | 44[41.9] | 61[58.1] | | | 63[60] | 42[40] | 105[67.3] | | |
| **Occupation** | | | | 2.41(1.52-3.82) | | | | 0.158 | 1.75(1.15-2.66) |
| Unemployed | 7[41.2] | 10[58.8] | 0.022 | | 10[58.8] | 7[41.2] | 17[10.9] | | |
| Student | 32[31.1] | 71[68.9] | | | 49[47.6] | 54[52.4] | 103[66.0] | | |
| Trader | 9[75] | 3[25] | | | 10[83.3] | 2[16.7] | 12[7.7] | | |
| Farmer | 1[100] | 0[0] | | | 0[0] | 1[100] | 1[0.6] | | |
| Civil servant | 8[42.1] | 11[57.9] | | | 11[57.9] | 8[42.1] | 19[12.2] | | |
| Artisan | 3[75.0] | 1[25] | | | 3[75.0] | 1[25] | 4[2.6] | | |
| **Marital status** | | | | | | | | | |
| Single | 47[34.8] | 88[65.2] | 0.031 | 1.66(1.07-2.57) | 67[49.6] | 68[50.4] | 135[86.5] | 0.062 | 1.49(1.02-2.18) |
| Married | 11[57.9] | 8[42.1] | | | 14[73.7] | 5[26.3] | 19[12.2] | | |
| Widowed | 2[100] | 0[0] | | | 2[100] | 0[0] | 2[1.3] | | |
| **Residence** | | | 0.755 | 1.15(0.84, 1.58) | | | | 0.373 | 0.96(0.71-1.30) |
| Urban | 29[35.8] | 52[64.2] | | | 44[54.3] | 37[45.7] | 81[51.9] | | |
| Rural | 31[41.3] | 44[58.7] | | | 39[52.0] | 36[48.0] | 75[48.1] | | |
| **Water supply** | | | | | | | | | |
| Sachet water | 40[40.8] | 58[59.2] | 0.465 | 1.20(0.87-1.66) | 52[53.1] | 46[46.9] | 98[62.9] | 0.532 | 0.99(0.74-1.32) |
| Pipe-borne | 19[33.9] | 37[66.1] | | | 30[53.6] | 26[46.4] | 56[35.9] | | |
| Borehole | 1[100] | 0[0] | | | 1[100] | 0[0] | 1[0.6] | | |
| Rain Water | 0[0] | 1[100] | | | 0[0] | 1[100] | 1[0.6] | | |
| **Walled residence** | | | | | | | | | |
| Yes | 23[35.4] | 42[64.6] | 0.534 | 1.15(0.84-1.58]) | 36[55.4] | 29[44.6] | 65[41.7] | 0.824 | 0.93(0.70-1.24) |
| No | 37[40.7] | 54[59.3] | | | 47[51.6] | 44[48.4] | 91[58.3] | | |
| **Cemented compound** | | | | | | | | | |
| Yes | 17[34.7] | 32[65.3] | 0.512 | 1.16(0.84-1.59) | 25[51] | 24[49] | 49[31.4] | 0.712 | 1.06(0.80-1.41) |
| No | 43[40.2] | 64[59.8] | | | 58[54.2] | 49[45.8] | 107[68.6] | | |
| **Cat ownership** | | | | | | | | | |
| Yes | 20[37.7] | 33[62.3] | 0.892 | 1.03(0.75-1.41) | 27[50.9] | 26[49.1] | 53[34.0] | 0.671 | 1.07(0.81-1.42) |
| No | 40[38.8] | 63[61.2] | | | 56[54.4] | 47[45.6] | 103[66.0] | | |
| **Eating undercooked meat** | | | | | | | | | |
| Yes | 37[41.6] | 52[58.4] | 0.373 | 1.21(0.88-1.67) | 53[59.6] | 36[40.4] | 89[57.1] | 0.213 | 1.33(0.97-1.84) |
| No | 23[34.3] | 44[65.7] | | | 30[44.8] | 37[55.2] | 67[42.9] | | |

Positive and negative results by sociodemographic and exposure factors stratified by T. gondii seropositivity (RDT and ELISA). Both conditional (stratified) and marginal (overall totals) distributions are shown. Prevalence ratios (PR), 95% confidence intervals (CI), and p values from Pearson's chi-square test (or Fisher's exact test where expected cell counts <5) are reported. PR = Prevalence Ratio; CI = Confidence Interval

**Table 5. Clinical characteristics and Blood Transfusion History of SCD patients stratified by laboratory diagnsis of T. gondii infection by RDT and ELISA.**

| Characteristics | RDT | | | | ELISA | | | | |
|---|---|---|---|---|---|---|---|---|---|
| | Positive | Negative | p-value | PR (95% CI) | Positive | Negative | Total, n [%] | p-value | PR (95% CI) |
| **Genotype** | | | | | | | | | |
| HbSS | 48[38.4] | 77[61.6] | 0.735 | 1.01(0.65-1.57) | 72[57.6] | 53[42.4] | 125[80.1] | 0.081 | 0.62(0.40-0.95) |
| HbSC | 12[38.7] | 19[61.3] | | | 11[35.5] | 20[64.5] | 31[19.9] | | |
| **Hydroxyurea Intake** | | | | | | | | | |
| Yes | 33[38.8] | 52[61.2] | 0.913 | 0.98(0.72-1.34) | 46[54.1] | 39[45.9] | 85[55.5] | 0.805 | 0.96(0.72-1.28) |
| No | 27[38.0] | 44[62.] | | | 37[52.1] | 34[47.9] | 71[45.5] | | |
| **Blood transfusion** | | | | | | | | | |
| Yes | 49[46.7] | 56[53.3] | 0.003 | 0.46(0.30-0.71) | 66[62.9] | 39[37.1] | 105[67.3] | 0.001 | 0.53(0.35-0.81) |
| No | 11[21.6] | 40[78.4] | | | 17[33.3] | 34[66.7] | 51[32.7] | | |
| **Reason of blood transfusion** | | | | | | | | | |
| Acute pain | 6[60] | 4[40] | 0.014 | 1.25(0.89-1.76) | 8[80] | 2[20] | 10[6.4] | 0.011 | 1.31(1.01-1.70) |
| Chronic anaemia | 35[50.7] | 34[49.3] | | | 42[60.9] | 27[39.1] | 69[44.2] | | |
| Acute pain+ Chronic Anaemia | 6[42.9] | 8[57.1] | | | 8[57.1] | 6[42.9] | 14[9.0] | | |
| Other complication | 1[16.7] | 5[83.3] | | | 5[83.3] | 1[16.7] | 6[3.9] | | |
| Unknown | 1[16.7] | 5[83.3] | | | 3[50] | 3[50] | 6[3.9] | | |

Results by genotype, hydroxyurea intake, transfusion history, and transfusion reasons stratified by T. gondii RDT and ELISA seropositivity. Both conditional (stratified) and marginal (overall totals) distributions are shown. Prevalence ratios (PR), 95% confidence intervals (CI), and p values from Pearson's chi-square test (or Fisher's exact test where expected cell counts <5) are shown. PR = Prevalence Ratio; CI = Confidence Interval.

**Table 6. Logistic regression for the odds of ELISA positive for _T. gondii_ among patients with SCD.**

| Characteristics | Unadjusted OR (95% CI) | P | Adjusted OR (95% CI) | P |
|---|---|---|---|---|
| **Age (years)** | | | | |
| ≤19 | 0.34(0.18-0.66) | 0.001 | 0.36(0.18-0.72) | 0.004 |
| ≥20 | 1 | | 1 | |
| **Blood Transfusion** | | | | |
| Yes | 3.39(1.67-6.84) | 0.001 | 3.39 (1.60-7.19) | 0.001 |
| No | 1 | | 1 | |
| **Education** | | | | |
| Basic (None/Primary/JHS) | 0.43(0.22-0.85) | 0.024 | 0.41(0.19-0.85) | 0.023 |
| Post Basic (SHS/Tertiary) | 1 | | 1 | |
| **Cat ownership** | | | | |
| No | 1.15(0.60-2.23) | 0.698 | 0.93(0.43-1.99) | 0.856 |
| yes | 1 | | 1 | |
| Consuming poorly cooked or raw meat | | | | |
| No | 0.55(0.29-1.05) | 0.063 | 0.47(0.23-0.98) | 0.041 |
| Yes | 1 | | 1 | |

Unadjusted and adjusted odds ratios (OR) with 95% confidence intervals (CI) and p-values are presented for age, blood transfusion, education, cat ownership, and consumption of poorly cooked or raw meat.

OR= Odds Ratio, CI= Confidence Interval.

| Diagnostics | Value |
|---|---|
| Sample Size | 156 |
| AUC (95% CI) | 0.722 (0.643-0.802) |
| Brier Score | 0.2088 |
| Brier Skill Score | 0.1615 |
| Hosmer-Lemeshow p-value | 0.4229 |
| McFadden's R² | 0.1239 |
| Nagelkerke R² | 0.2101 |
| Mean variance inflation factor (VIF) | 1.17 |

### Stratum-specific odds ratios for the age × transfusion Interaction

In the full multiple logistic regression model, the age × transfusion interaction term was statistically significant (p = 0.02). Stratum-specific odds ratios indicated that among SCD patients ≤19 years, transfusion was associated with 2.30 times higher odds of *T. gondii* seropositivity (95% CI: 0.94–5.62; p = 0.07). In contrast, among SCD patients ≥20 years, transfusion was associated with 4.29 times higher odds of seropositivity (95% CI: 1.34–13.74; p = 0.01).

## Discussion

Scientific literature is replete with several studies reporting varied prevalence rates of *T. gondii* infection in different populations [18,20–23]. Few studies have reported the prevalence in SCD patients [13,14], and to our knowledge this present study is the only one that has reported the prevalence of *T. gondii* infection among SCD in sub-saharan Africa. The ELISA-based seroprevalence of *T. gondii* in our study (53.2%) exceeded the 45% reported among SCD patients in Brazil [13], suggesting possible regional differences in exposure risk, screening practices, or environmental factors.. However because of the reported high prevalence of *T. gondii* infections in Ghana, our finding is somewhat moderate compared with studies done among other populations such as pregnant women, HIV patients, children and other volunteers, where higher prevalence of above 80% has been reported. The relatively smaller sample size of our study might explain the lower prevalence compared to other studies in Ghana with higher prevalence.

There was an obvious discrepancy between the results of RDT and ELISA seen in this study, with a higher seropositivity for ELISA compared to RDT, a finding corroborated by other sero-prevalence studies [24–26]. It is important to note that while ELISA is a validated and a strong reference standard compared to RDT, it is not a perfect gold standard as there are false results reported coupled with insensitivity issues concerning discriminating between acute and chronic toxoplasmosis [25]. Although RDTs are generally less sensitive than ELISA, the RDT used in our study demonstrated comparatively strong performance, with a sensitivity of 68.7%, specificity of 95.9%, PPV of 95%, NPV of 72.9%, diagnostic efficiency of 81.4% and Cohen's kappa of 0.634. These values were markedly superior to those reported in earlier seroprevalence studies. For example, Bassiony et al., (2016) [24] documented an RDT sensitivity of only 15.8%, with a diagnostic efficiency of 49.2% and NPV of 41.5%, while Singh et al., (2021) [26] reported a sensitivity of 31%, specificity of 91.2%, and Cohen's kappa of 0.20. These differences highlight the variability in diagnostic performance across RDT brands, and the kit employed in our study appears to belong to the subset with comparatively higher accuracy [26]. However since predictive values (PPV and NPV) depend on disease prevalence, the high PPV (95%) and moderate NPV (72.9%) for RTD observed in our study are likely explained by the relatively high prevalence of ELISA *T. gondii* seropositivity of 53%. Conversely in lower-prevalence populations, PPV would decline while NPV would improve, thus underscoring that the utility of RDT varies with epidemiological context and predictive values should be interpreted cautiously outside the study population.

A significant association was observed between patient age and *T. gondii* seropositivity for both RDT (p ≤ 0.001) and ELISA (p = 0.003), with higher proportion of *T. gondii* seropositivity consistently detected among those aged 20 years and

above. This finding is in agreement with Ferreira et al., (2017) [13], where seronegativity was seen more among paeditric patients with SCD. More cumulative exposure to varied risk factors of *T. gondii* transmission among older SCD patients might be responsible for this finding [13]. However this was in contrast with a study conducted by Ayi et al., (2016) [27]who reported a rather high prevalence among children in accra Ghana.

Barely a third (33.9%) of the SCD participants affirmed to having cats has pets which is a very common practice in Ghana second only to dog ownership [28]. There was no significant association between cat ownership and *T. gondii* seropositivity for RDT or ELISA in this study. Despite it is an established fact that cats fuel the transmission of *T. gondii* in humans and animals, cat ownership does not invariably increase the risk of *T. gondii* infection [27,29,30]. However some other studies identified increased risk of infection associated with cat ownership [31].

About 56% of participants in this study agreed to have or are eating raw or undercooked meat. While the average Ghanaian consumes meat daily, consumption of raw meat is not culturally entrenched [32,33]. Human infection with *T. gondii* is heavily linked with the consumption of undercooked or raw meat [23]. Again this risk factor did not show any significant association with *T. gondii* seropositivity for either RDT or ELISA, a finding consistent with report from some other studies [24,27]. However another study in Egypt reported an increased odds of *T. gondii* infections among people who consumed raw meat [34].

Unlike cat ownership, consumption of raw meat and other risk factors, blood transfusion as a risk factor for *T. gondii* infection might be restricted to populations with certain medical condition that requires blood transfusion as part of the therapeutic intervention and management. SCD is one of such medical condition that requires frequent blood transfusion [16,35,36].This was supported by our study with more than 65% of the SCD patients affirmed to have had blood transfusion in their lifetime. Clinical reasons for blood transfusions among SCD patients are diverse, and include severe anaemia, vasco-oclusive crises (VOC), acute chest syndrome, stroke and others [37,38]. In our study, severe anaemia was the most common reason followed by VOC.

Blood transfusion was significantly associated with both RDT and ELISA seropositivity. SCD patients with blood transfusion had 3 times the odds of testing positive with ELISA (adjusted OR 3.14[95% CI 1.50–6.58]; p = 0.002). Also, when the age × transfusion interaction was examined, transfusion increased odds of *T. gondii* seropositivity in both age strata **of ≤19 and ≥20 years**, but the effect was stronger among adults with SCD (≥20 years: OR = 4.29, 95% CI: 1.34–13.74; p = 0.01) compared to younger SCD patients (≤19 years: OR = 2.30, 95% CI: 0.94–5.62; p = 0.07). This finding suggests that transfusion-related risk is present across age groups but more pronounced among adults, possibly reflecting cumulative exposure, differences in transfusion practices, or immunological variation [13,16,35,36]. This study was in contrast with some studies which did not find any significant association between positive ELISA test results and blood transfusion among SCD patients [13,14]. Blood transfusion transmission of *T. gondii* may not be merely a theoretical possibility but a genuine public health concern given that an estimated 33% of blood donors globally and 46% in are infected with *T. gondii* [39].

However the presence of *T. gondii* anti-IgM or IgG does not necessarily mean the presence of the parasite in the blood or active infection, for the risk of transmission is higher in the presence of parasites (tachyzoites) usually seen in blood donors with acute infection [10,40,41]. Although IgM and IgG suggest acute and chronic infection respectively, anti-*T. gondii.* IgG and IgM can be seen in both acute and chronic infections [10,41]. The unreliability to accurately determine acute or chronic infections among blood donors using serology probably suggest either a more superior screening tool such as PCR be employed in detecting the presence of the parasite among blood donors [12,40,42,43], or to exclude blood donors with anti-*T. gondii* IgG or IgM from donating especially to the immunocompromised and pregnant women [12]. The latter might be an easier route to follow especially in resource limited settings where more sensitive molecular techniques for screening are logistically difficult to achieve [44].

Taken together, the lack of routine donor screening for *T. gondii* may have important public health implications, particularly in resource-limited settings such as Ghana where blood transfusion remains a critical component of SCD

management. Strengthening transfusion safety protocols would not only protect high-risk recipients but also reinforce national blood safety standards. These findings underscore the need for policymakers and health authorities to consider incorporating *T. gondii* screening into comprehensive donor evaluation strategies in Ghana.

This study reported HbSS (79.5%) and HBSC(20.5%) among the SCD patients in this study, a finding that is in agreement with the national report where HbSS is the prevalent variant of SCD [19,45]. Thalassemia hemoglobinopathy was not reported among the SCD patients in this study which might be due to the limitation of the commonly used, less sensitive Hb electrophoresis diagnostic method used in this study [46–48]. Thus, it is possible that there were co-occurrence of HbSS or HbSC with thalassemia in this study, but the Hb **electrophoresis** was not sensitive enough to make the diagnosis.

It is important to state some limitations of this study. Firstly the crosssectional study design of this study can not categorically establish that blood transfusion is a driver of *T. gondii* seropositivity among the SCD participants, for the significant association observed might be confounded by age and cumulative community exposure in a high-prevalence setting. Sensitivity analyses, such as stratification by exposure, would have strengthened the robustness of this finding; however, the limited sample size of our study was a constraint. Therefore a more robust longitudinal study design with a larger sample size will be better in establishing temporality and show more clearly the role of blood transfusion in *T. gondii* seropositivity. Secondly this was a hospital based study from the small population of registered SCD patients in the clinic, which might not be representative of the SCD patient in the region. Caution therefore, should be applied in generalizing the findings of this study.

Despite all these limitations, the findings of this study reported for the first in Ghana, have shed some light on the possible role of blood transfusion in *T. gondii* infection among patients with SCD in the region and call for more detailed studies to elucidate the contribution of blood transfusion transmission of *T. gondii*.

## Conclusion

This study on *T. gondii* prevalence among SCD patients in Ho, Ghana, found moderate rates of 38.5% (RDT) and 53.2% (ELISA), with higher detection via ELISA. Age and blood transfusion history were significantly linked to infection, whereas common exposures such as cat ownership and undercooked meat showed no significant association. The strong association between *T. gondii* seropositivity and blood transfusion observed in this study might have been confounded by age and cumulative exposure to infection through other risk factors. Thus, more robust longitudinal or donor-recipient studies are needed to help establish the true contribution of blood transfusion in the transmision of *T. gondii* in the population especially among frequent recipients of blood transfusion such as patients with SCD

## Supporting information

**S1 Questionnaire. QUESTIONNAIRE_TOXO_SCD.**
(DOCX)

**S2 Data. Minimal anonymized data set.**
(XLSX)

## Acknowledgments

We want to thank all the staff of the Sickle Cell Disease clinic, the laboratory staff for making this study possible. More especially we want to thank the participants who willingly took part in this study. Thank you all for your enthusiastic collaboration, your participation is truly appreciated.

## Author contributions

**Conceptualization:** verner N. orish.

**Data curation:** verner N. orish, David Adzah.

**Formal analysis:** verner N. orish.

**Investigation:** Renosten E. Tetteh, Chinecherem A. Ndiokwelu, Emmanuel A. Allotey, Evans A. Yeboah.

**Methodology:** Renosten E. Tetteh, David Adzah.

**Project administration:** David Adzah, Chinecherem A. Ndiokwelu, Emmanuel A. Allotey, Evans A. Yeboah.

**Writing – original draft:** verner N. orish, Renosten E. Tetteh.

**Writing – review & editing:** David Adzah, Chinecherem A. Ndiokwelu, Emmanuel A. Allotey, Evans A. Yeboah, Sylvester Y. Lokpo, Kenneth Ablordey, Duneeh R. Vikpebah, Ekene K. Nwaefuna, Precious K. Kwadzokpui, Noble D. Dika, Elom Y. Dzefi, Kokou H. Amegan-Aho, Aninagyei Enoch, Senyo Tagboto.

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
