## [Decision Letter · Decision Letter 0]

4 Nov 2025

Dear Dr. orish,

We look forward to receiving your revised manuscript.

Kind regards,

Masoud Foroutan, Ph.D.

Academic Editor

PLOS ONE

Journal Requirements:

Reviewers' comments:

Reviewer's Responses to Questions

**Comments to the Author**

1. Is the manuscript technically sound, and do the data support the conclusions?

Reviewer #1: Yes

Reviewer #2: Partly

Reviewer #3: Partly

2. Has the statistical analysis been performed appropriately and rigorously?

Reviewer #1: No

Reviewer #2: I Don't Know

Reviewer #3: Yes

3. Have the authors made all data underlying the findings in their manuscript fully available?

Reviewer #1: Yes

Reviewer #2: Yes

Reviewer #3: Yes

4. Is the manuscript presented in an intelligible fashion and written in standard English?

Reviewer #1: Yes

Reviewer #2: Yes

Reviewer #3: No

Reviewer #1: 1. Methods should clearly state which test (ELISA vs RDT) defines seropositivity for the association analyses. If ELISA is the reference standard, all inferential analyses (logistic regression) should use ELISA-defined seropositivity. Currently Tables and text sometimes mix RDT and ELISA and this is confusing.

2. The authors imply that past transfusions lead to seropositivity. This is not provable in a cross-sectional design because timing of infection relative to transfusion is unknown. Reword conclusions and discuss that transfusion is an exposure that is consistent with but not proven to be the transmission route.

3. Define precisely: “ever had blood transfusion” vs “transfusion in past 12 months” (both appear in the manuscript). Use one primary exposure and justify. If both are used, show separate models.

4. For participants with transfusion in the past 12 months, consider sensitivity analysis excluding individuals whose seropositivity might have preceded transfusion (if dates available) or at least stratify by time since last transfusion.

5. For every table and subgroup report the exact n used. If any missingness exists for covariates or outcomes, report number and pattern; explain handling (complete-case only, imputation).

6. Table 4 & 5: For subgroup comparisons (age categories, genotype, etc.) provide risk differences and 95% CIs or prevalence ratios rather than p-values alone. This gives clinical meaning to findings.

7. Use Fisher’s exact test when expected cell counts <5. For example, groups with very small n (age ≥60) must use Fisher’s exact or combine categories.

9. Provide a priori rationale for covariates included in the adjusted model. Avoid data-driven “stepwise” selection unless justified; if used, report selection procedure and results.

10. For each fitted logistic model, report:

• Number of events and non-events (EPV — events per variable) to ensure model is not overfitted. With 83 ELISA positives, be cautious about the number of covariates in adjusted model (rule of thumb: >10 events per parameter).

• VIFs for multicollinearity.

• Hosmer–Lemeshow result

• ROC AUC with 95% CI.

11. Investigate interaction between age and blood transfusion (biologically plausible). If interaction is present, present stratified ORs.

13. Provide the full 2×2 table in the main text or supplement (you have provided TP/TN/FP/FN but ensure clarity). State whether ELISA is true gold standard or imperfect reference and discuss implications.

14. Report exact binomial (Clopper–Pearson) CIs for sensitivity/specificity and predictive values; indicate the method used to compute CIs.

15. Kappa = 0.634 is “substantial” agreement; state the scale used for interpretation (e.g., Landis & Koch).

16. Discuss that PPV/NPV depend on prevalence and provide context (e.g., expected values in general population).

Reviewer #2: Dear Editor,

In a cross-sectional study conducted with 156 patients suffering from sickle cell disease (SCD) at Ho Teaching Hospital in the Volta Region of Ghana, the authors observed that the seroprevalence of Toxoplasma gondii, detected by ELISA, was significantly higher at 53.0% compared to 38.5% detected by rapid diagnostic tests (RDT). Furthermore, a history of blood transfusion was associated with an increased likelihood of T. gondii seropositivity among SCD patients, suggesting that transfusions may represent a potential route for transmission within this population.

Several questions arise from this study:

- Were consecutive SCD (sickle cell disease) patients enrolled, or is there potential for sampling bias?

- Were any exclusion criteria applied that might influence seroprevalence?

- How were discordant IgG and IgM results managed?

- A positive IgG result indicates past exposure to the infection but does not confirm an active infection. To distinguish the timing of infection, IgG avidity testing should be performed.

- Why didn't you use molecular tests to detect active infections?

- How was transfusion history defined in this study?

- Were potential confounders included in the analysis model?

- What is the prevalence of Toxoplasma gondii in the general population?

- How is transfusion interpreted as a transmission route, given the cross-sectional design?

- How is the discrepancy between ELISA and RDT prevalence reconciled?

- Were the results communicated to the clinic or to the patients, and was follow-up offered for those who tested positive?

Reviewer #3: Dear editor

Thank you for the opportunity to review the manuscript titled “seroprevalence of Toxoplasma gondii among sickle cell disease (SCD) patients in Ghana”. I would like to appreciate the authors' effort to undertake this valuable task”. The study investigates the seroprevalence of Toxoplasma gondii among sickle cell disease (SCD) patients in Ghana, with emphasis on the potential association between blood transfusion history and seropositivity. The topic is clinically relevant and fills a local research gap, especially since transfusion-transmitted T. gondii infections are poorly characterized in sub-Saharan Africa.

However, despite its importance, the study has several methodological and presentation flaws that undermine the robustness and reproducibility of the findings. Major revisions are required before the paper can be considered for publication in a journal such as PLOS ONE.

Major Comments

Study Design and Objectives

The cross-sectional design is appropriate for prevalence estimation but cannot establish causality or temporal relationships between transfusion and infection. The authors acknowledge this in the discussion, but they should emphasize it clearly in the Abstract and Methodology.

The objective (“to find out the prevalence of T. gondii infection and the association of blood transfusion among patients with SCD”) is acceptable but could be phrased more precisely as:

“To determine the prevalence of anti-Toxoplasma gondii antibodies among SCD patients and to assess the association between seropositivity and history of blood transfusion.”

Sample Size and Sampling Procedure

The sample size calculation using Cochran’s formula is correctly presented, but the population (N=300) was taken from clinic attendance, not total SCD population in the region—this introduces selection bias. The manuscript lacks details on sampling technique (e.g., consecutive sampling, random selection). Specify how participants were recruited to ensure representativeness.

Diagnostic Testing

The use of RDT and ELISA is appropriate, but the RDT performance characteristics should have been validated with a local control group or WHO-standard panels to confirm accuracy.

Details on kit lot numbers, manufacturer references, and quality control measures are missing. For reproducibility, these must be stated.

Defining “seropositive” as IgG or IgM or both is acceptable, but distinguishing recent (IgM) from past (IgG) infections would have improved epidemiological interpretation.

Data Analysis

The analysis using SPSS v25 is fine, but the variable coding and handling of missing data are not described.

The logistic regression model should include confidence intervals for all variables, not only blood transfusion. Also, justification for including variables in the multivariate model should be given (based on univariate p<0.2, for example).

The adjusted OR (3.14) is significant but could be confounded by age and exposure factors not properly adjusted for. Sensitivity analyses would strengthen the claim.

Results Interpretation

The results are clearly tabulated, but some tables are overcrowded (e.g., Table 4 and 5) — consider splitting into demographic vs. clinical characteristics.

The conclusion that blood transfusion may transmit T. gondii is overstated. Seropositivity does not confirm transmission via transfusion; longitudinal or donor-recipient studies are needed.

Discussion should better distinguish between association and causation, and avoid speculative statements without molecular evidence.

Language and Style

Numerous grammatical, typographical, and stylistic errors (e.g., “varried,” “haemoglobinopathy,” “lectrophoresis”) need correction by a professional English editor.

The tone should be more scientific and concise; many sentences in the Introduction and Discussion are repetitive or verbose.

References are inconsistently formatted — ensure compliance with PLOS ONE reference style (DOI, complete author list, year, journal, volume, pages).

Ethical and Data Availability Statements

Data availability statement (“data cannot be shared publicly because it was generated from patients attending the Ho Teaching Hospital”) does not meet PLOS ONE’s open data policy. A controlled-access mechanism (e.g., through institutional data office) should be mentioned.

Minor Comments

Abstract:

Include numeric results for odds ratios and p-values.

Avoid abbreviations like “RDT” without prior definition.

Figures:

Figures 1–3 are informative but should include error bars and legends. The quality should be improved for publication.

Tables:

Add column totals and indicate statistical tests (e.g., χ², Fisher’s exact) in captions.

For clarity, present p-values to three decimal places consistently.

Introduction:

Reduce literature overload; several global prevalence citations could be summarized more succinctly.

Add rationale for selecting SCD patients beyond frequent transfusion — e.g., immunocompromised state.

Discussion:

Avoid restating the results; instead, focus on comparative analysis with prior studies.

Include a short section on public health implications, e.g., the need for donor screening for T. gondii in Ghana.

Conclusion:

Should emphasize limitations and recommend future longitudinal or molecular studies.

.

Reviewer #1: No

Reviewer #2: No

Reviewer #3: No

---

## [Author Response · Author response to Decision Letter 1]

4 Dec 2025

Response to Reviewers

We the authors are very grateful to the reviewers of our manuscript. We acknowledge the effort and time put into this review to make our study better,

Find attached comments to all the issues raised.

---

## [Decision Letter · Decision Letter 1]

28 Jan 2026

Dear Dr. orish,

Thank you for submitting your manuscript to PLOS ONE. After careful consideration, we feel that it has merit but does not fully meet PLOS ONE’s publication criteria as it currently stands. Therefore, we invite you to submit a revised version of the manuscript that addresses the points raised during the review process.

We look forward to receiving your revised manuscript.

Kind regards,

Masoud Foroutan, PhD

Academic Editor

PLOS One

Journal Requirements:

Reviewers' comments:

Reviewer's Responses to Questions

**Comments to the Author**

Reviewer #1: (No Response)

Reviewer #4: (No Response)

2. Is the manuscript technically sound, and do the data support the conclusions?

Reviewer #1: (No Response)

Reviewer #4: Partly

3. Has the statistical analysis been performed appropriately and rigorously?

Reviewer #1: (No Response)

Reviewer #4: Yes

4. Have the authors made all data underlying the findings in their manuscript fully available?

Reviewer #1: (No Response)

Reviewer #4: Yes

5. Is the manuscript presented in an intelligible fashion and written in standard English?

Reviewer #1: (No Response)

Reviewer #4: Yes

Reviewer #1: • The primary outcome should be clearly defined as ELISA-defined seropositivity. Analyses using RDT results should be limited to reporting diagnostic performance and should not be used as the outcome in regression models to avoid misinterpretation due to misclassification. State the seropositivity definition in the Methods section and reference the ELISA cutoff per manufacturer; reserve RDT results for supplementary sensitivity analyses.

• Choose a single primary exposure (e.g., ever transfused or transfusion in the past 12 months) with a priori justification.

• Present full diagnostic metrics for the adjusted model: variance inflation factors (VIFs) for multicollinearity, Hosmer-Lemeshow goodness-of-fit, area under the curve (AUC) with 95% confidence interval, and calibration metrics (e.g., Brier score or calibration plot).

• Interaction between age and transfusion was considered; the authors decided not to pursue it due to a non-significant p-value in the figure. Pre-specify a clinically plausible interaction (e.g., age × transfusion) and test it in the full model. Report stratum-specific odds ratios if interaction is present. If not, state that the interaction was not supported by the data and discuss clinical relevance.

• Ensure tables (Table 1 and Table 2) include column totals where helpful, with explicit test names in captions.

• Throughout the text, use "multiple logistic regression" instead of "multivariate logistic regression."

Reviewer #4: Title: Toxoplasma gondii seropositivity among patients with sickle cell disease: Prevalence and Association with Blood Transfusion History

Comment to authors

General comment

Authors dedicate the comments of all reviewers but some are still need to get more explanation. Title is more prefer to the epidemiological study but the methods are combined with diagnostic accuracy study. The reviewers also give the comment for confusing the usage both RTD and ELISA for assessing the seropositive [already known that the ELISA is more accurate for detection than the RTD – to align with the title, just suggest 1- removing the result of accuracy finding, 2- all analysis with reference diagnosis (ELISA) by setting the operational definition of seropositive.]

Specific comment

Comment 1: Abstract, p value should be consistent in decimal (suggested three decimal)

Comment 2: Abstract, the sentence of “This cross-sectional design allowed estimation of prevalence and associations, but not causality or temporality. More robust longitudinal studies are needed to help elucidate the true contribution of blood transfusion transmission of T. gondii” should be removed to limitation in discussion.

Comment 3: Abstract, the recommendations based on significant associations should be added in conclusion.

Comment 4: Study Design, “study design was employed to estimate the prevalence of T. gondii infection and to examine associations with blood transfusion history. While suitable for prevalence estimation and association testing, this design cannot establish causal or temporal relationships.” Is nearly the same with Line No. 105-107 and these sentences should be removed.

Comment 5: Study Area and Site, the subheading should be “Study Area”.

Comment 6: Sample size determination, please recheck the sample size (would be 169?).

Comment 7: Sample size determination, can you explain the usage of n = n/ 1+(n/N) instead of n = n/ 1+(n-1/N) as Cochran’s modified formula for finite populations?

Comment 8: Sampling and Recruitment Procedure, please revise as “Given the relatively small clinic population of 300 registered patients, a consecutive sampling was employed, where all eligible SCD patients attending the clinic during the study period were recruited sequentially until the required sample size was achieved” in Line No. 162-165.

Comment 9: Figure 1 and 2, please specify visualization to clear for describing the type of variables. Error bars are reliable for numerical data and should not be add in description of categorical or nominal data. (numerical data - bar for mean and error bar for its SD or SE, categorical or nominal data - bar for number or percent and error bar for 95% CI of percent).

.

Reviewer #1: No

Reviewer #4: No

---

## [Author Response · Author response to Decision Letter 2]

17 Feb 2026

RESPONSE TO REVIEWERS

We sincerely thank the reviewers for their painstaking effort in improving our manuscript through valuable comments and constructive suggestions. We have carefully considered each point raised and provide detailed responses below. In revising the manuscript, we have incorporated the reviewers’ insights to enhance clarity, strengthen the methodology, and improve the overall quality of the paper.

Below, we present our responses to the specific issues raised:

Reviewer #1:

• The primary outcome should be clearly defined as ELISA-defined seropositivity.

Response: this the main focus of the study. It has now been clearly stated in the manuscript. See pages # 12 line # 289-290 “For this study, seropositivity was operationally defined using ELISA results, and this served as the primary outcome of interest”.

Analyses using RDT results should be limited to reporting diagnostic performance and should not be used as the outcome in regression models to avoid misinterpretation due to misclassification.

Response: ELISA was used as the primary outcome of regression model and not RDT. See data analysis pages # 13 line # 311-312 “Multiple logistic regression was performed to estimate the association between blood transfusion and the odds of T. gondii ELISA seropositivity” and result pages # 27 line # 498-499 “Multiple logistic regression analysis was further carried out on variables with significant association with ELISA resuts to find out the odds of positive ELISA results among the SCD ptients”

State the seropositivity definition in the Methods section and reference the ELISA cutoff per manufacturer; reserve RDT results for supplementary sensitivity analyses.

Response: this has been stated See pages # 12 line # 284-285 “To determine the critical cut-off, the average absorbance of the negative controls was summed with 0.15; page # 12 line # 289-290 “ELISA positive is defined as positive for either IgM, IgG or IgM and IgG combination.”

• Choose a single primary exposure (e.g., ever transfused or transfusion in the past 12 months) with a priori justification.

Response: This has been done. “Transfused in the past 12 month” was considered to be redundant and has been appropriately deleted See pages # 9 line # 226-228 “Information on the lifetime history of blood transfusion was obtained. This was defined as having ever received one or more transfusions of any type, with responses categorized as “Yes” or “No.”.

• Present full diagnostic metrics for the adjusted model: variance inflation factors (VIFs) for multicollinearity, Hosmer-Lemeshow goodness-of-fit, area under the curve (AUC) with 95% confidence interval, and calibration metrics (e.g., Brier score or calibration plot).

Response: We thank the reviewer for the valuable suggestion to present full diagnostic metrics for the adjusted prediction model. In the revised manuscript, we have now included comprehensive model validation results and added all requested diagnostics. See page # 28 line # 512-522

• Interaction between age and transfusion was considered; the authors decided not to pursue it due to a non-significant p-value in the figure. Pre-specify a clinically plausible interaction (e.g., age × transfusion) and test it in the full model. Report stratum-specific odds ratios if interaction is present. If not, state that the interaction was not supported by the data and discuss clinical relevance.

Response: We appreciate the reviewer’s comment. In the revised manuscript, we have clarified that a clinically plausible interaction between age and transfusion was pre specified and tested in the full multivariate model. The interaction term was statistically significant (p = 0.02), and stratum specific odds ratios were therefore reported see method/analysis section page # 13, line # 327-330; result section page # 29 line # 536-541; discussion page 32 line # 626-632 52

• Ensure tables (Table 1 and Table 2) include column totals where helpful, with explicit test names in captions.

Response: columns total has been added to table 1 and 2. These tables summarize the background profile of the SCD patients (age, sex, education, genotype, transfusion history, etc.) but they do not present diagnostic test results (RDT or ELISA outcomes)

• Throughout the text, use "multiple logistic regression" instead of "multivariate logistic regression."

Response: this has been done, thank you very much

Reviewer #4: Title: Toxoplasma gondii seropositivity among patients with sickle cell disease: Prevalence and Association with Blood Transfusion History

Comment to authors

General comment

Authors dedicate the comments of all reviewers but some are still need to get more explanation. Title is more prefer to the epidemiological study but the methods are combined with diagnostic accuracy study. The reviewers also give the comment for confusing the usage both RTD and ELISA for assessing the seropositive [already known that the ELISA is more accurate for detection than the RTD – to align with the title, just suggest 1- removing the result of accuracy finding,

2- all analysis with reference diagnosis (ELISA) by setting the operational definition of seropositive.]

Response: We appreciate the reviewer’s comment regarding the alignment of the title with the study scope. We have retained the original title, “Toxoplasma gondii seropositivity among patients with sickle cell disease: Prevalence and Association with Blood Transfusion History,” because the primary focus of the study is epidemiological — estimating prevalence and examining associations with transfusion history.

We also included RDT results to evaluate diagnostic accuracy against ELISA, given the widespread use of RDTs in resource limited settings. This secondary analysis broadens the relevance of the study by providing practical insights into diagnostic performance, while not altering the epidemiological focus signaled in the title. The Methods and Results are aligned to make this distinction explicit, ensuring clarity for readers.

Clearly stated in the Methods, ELISA defined seropositivity was the operational definition and the primary outcome for all prevalence and association analyses.

“For this study, seropositivity was operationally defined using ELISA results, and this served as the primary outcome of interest.. (Methods, ELISA procedure See pages # 12 line # 289-290)

“The diagnostic performance of the rapid diagnostic test (RDT) was evaluated against ELISA as the reference standard Methods”, (Data Analysis page # 12 line # 302-303… Multiple logistic regression was performed to estimate the association between blood transfusion and the odds of T. gondii ELISA seropositivity.” ( Methods, Data Analysis page # 13 line 313-314).

Specific comment

Comment 1: Abstract, p value should be consistent in decimal (suggested three decimal)

RESPONE: This has been done thank you very much

Comment 2: Abstract, the sentence of “This cross-sectional design allowed estimation of prevalence and associations, but not causality or temporality. More robust longitudinal studies are needed to help elucidate the true contribution of blood transfusion transmission of T. gondii” should be removed to limitation in discussion.

Response: We appreciate the reviewer’s comment regarding the placement of the limitation sentence. While we acknowledge the suggestion, this statement is already in the limitation section of discussion (see the highlighted texts on page # 34 line # 663-678).

However, we have chosen to retain it in the Abstract in agreement with a previous reviewer’s recommendation, but revised it into a concise sentence. We believe that acknowledging this limitation at the Abstract stage not only provides readers with immediate clarity on the scope of the study but also constructively emphasizes the need for future longitudinal or interventional studies to elucidate the role of blood transfusion.

Comment 3: Abstract, the recommendations based on significant associations should be added in conclusion.

Response: We thank the reviewer for this valuable suggestion. In line with the significant associations observed in our study, we have revised the Abstract conclusion to include clear recommendations.

Specifically, we now highlight that the strong association between T. gondii seropositivity and blood transfusion underscores the need to strengthen transfusion safety protocols and consider screening strategies for T. gondii among high risk populations such as patients with sickle cell disease.

Comment 4: Study Design, “study design was employed to estimate the prevalence of T. gondii infection and to examine associations with blood transfusion history. While suitable for prevalence estimation and association testing, this design cannot establish causal or temporal relationships.” Is nearly the same with Line No. 105-107 and these sentences should be removed.

Response: We thank the reviewer for this observation. We agree that the sentences are redundant and they have been deleted. Thank you very much.

Comment 5: Study Area and Site, the subheading should be “Study Area”.

Response: Thank you very much, this has been done

Comment 6: Sample size determination, please recheck the sample size (would be 169?).

Response: We thank the reviewer for pointing out this issue. On re checking our calculation, we realized that the population size was incorrectly stated as 300 in the initial draft. The correct number of registered sickle cell disease patients at the clinic is 210. Using Cochran’s formula with p=0.5, Z=1.96, and allowable error of 0.05, the initial sample size was 385. Applying the finite population correction for N=210 gave a minimum sample size of 136. To increase the power of the study, we recruited 156 patients, which exceeds the minimum required. We have corrected this section in the Methods to reflect the accurate calculation and population size.

Comment 7: Sample size determination, can you explain the usage of n = n/ 1+(n/N) instead of n = n/ 1+(n-1/N) as Cochran’s modified formula for finite populations?

Response: thank you for raising this important point. In our initial draft, we presented the finite population correction formula in a simplified form as n=n_0/(1+n_0/N). We acknowledge that the precise Cochran’s modified formula is n=n_0/(1+(n_0-1)/N). The difference between these two versions is negligible when n_0is large, as in our case (n_0=385), but for rigor and clarity we have corrected the Methods section to reflect the exact formula. n=n_0/(1+(n_0-1)/N).

Using the corrected formula with N=210 registered sickle cell disease patients, the minimum sample size is 136. We ultimately recruited 156 patients, which exceeds the minimum required and ensures adequate study power. We have revised the Methods section accordingly to present the correct formula and calculation step by step.

Comment 8: Sampling and Recruitment Procedure, please revise as “Given the relatively small clinic population of 300 registered patients, a consecutive sampling was employed, where all eligible SCD patients attending the clinic during the study period were recruited sequentially until the required sample size was achieved” in Line No. 162-165.

Response: Thank you very much for this suggestion. This has been revised as instructed

Comment 9: Figure 1 and 2, please specify visualization to clear for describing the type of variables. Error bars are reliable for numerical data and should not be add in description of categorical or nominal data. (numerical data - bar for mean and error bar for its SD or SE, categorical or nominal data - bar for number or percent and error bar for 95% CI of percent).

Response: Thank you for highlighting the need to align error bars with the type of variable. In the revised manuscript, Figures 1 and 2 have been updated to ensure clarity and consistency with statistical conventions. Specifically, both figures present categorical variables (blood transfusion distribution and risk factors). Accordingly, the bar charts now display percentages with error bars representing the 95% confidence intervals of those percentages, rather than standard errors of the mean.

---

## [Decision Letter · Decision Letter 2]

26 Feb 2026

Dear Dr. orish,

We look forward to receiving your revised manuscript.

Kind regards,

Masoud Foroutan

Academic Editor

PLOS One

Journal Requirements:

Reviewers' comments:

Reviewer's Responses to Questions

**Comments to the Author**

Reviewer #1: (No Response)

Reviewer #4: All comments have been addressed

2. Is the manuscript technically sound, and do the data support the conclusions?

Reviewer #1: Partly

Reviewer #4: Yes

3. Has the statistical analysis been performed appropriately and rigorously?

Reviewer #1: No

Reviewer #4: Yes

4. Have the authors made all data underlying the findings in their manuscript fully available?

Reviewer #1: Yes

Reviewer #4: Yes

5. Is the manuscript presented in an intelligible fashion and written in standard English?

Reviewer #1: Yes

Reviewer #4: Yes

Reviewer #1: The suggested revisions to Table 1 and Table 2 are intended to improve the clarity and interpretability of the data distributions. Specifically:

A total column should be added to summarize the overall frequency or percentage across all categories for each explanatory variable.

Two separate columns should be included to show the distribution of each explanatory variable stratified by the levels of the outcome (response) variable.

This structure will allow readers to clearly see both:

(a) the conditional distribution of each predictor across outcome strata, and

(b) the marginal (overall) distribution of each predictor in the entire sample.

Furthermore, it is strongly recommended that:

Column totals be included where they meaningfully contribute to data interpretation.

The exact names of the statistical tests used (e.g., chi-square test, Fisher's exact test, trend test) be explicitly stated in the table captions to ensure transparency and reproducibility.

Reviewer #4: (No Response)

.

Reviewer #1: No

Reviewer #4: No

---

## [Author Response · Author response to Decision Letter 3]

9 Mar 2026

RESPONSE TO REVIEWERS

We the author once again want to thank the editor and reviewers for the guidance so far. Thank you very much for the comments. kindly find below response to the issues raised.

Reviewer #1: The suggested revisions to Table 1 and Table 2 are intended to improve the clarity and interpretability of the data distributions. Specifically:

A total column should be added to summarize the overall frequency or percentage across all categories for each explanatory variable.

Two separate columns should be included to show the distribution of each explanatory variable stratified by the levels of the outcome (response) variable.

This structure will allow readers to clearly see both:

(a) the conditional distribution of each predictor across outcome strata, and

(b) the marginal (overall) distribution of each predictor in the entire sample.

Furthermore, it is strongly recommended that:

Column totals be included where they meaningfully contribute to data interpretation.

The exact names of the statistical tests used (e.g., chi-square test, Fisher's exact test, trend test) be explicitly stated in the table captions to ensure transparency and reproducibility.

Response to Reviewer #1:

We appreciate the reviewer’s thoughtful recommendations to restructure Tables 1 and 2 by adding total columns and stratified columns for outcome variables. While we fully agree with the importance of clarity and interpretability, we respectfully propose to retain Tables 1 and 2 in their current format as descriptive summaries of the overall (marginal) distributions of sociodemographic and clinical characteristics. This approach allows these tables to serve as baseline descriptive data for the study population. However, table 4 and 5 provided some of the information

To address the concern, we have:

1. Added total columns to Tables 4 and 5 to summarize the overall frequency and percentage across categories, thereby improving clarity.

2. Directed readers to Tables 4 and 5, which already present the stratified (conditional) distributions by T. gondii RDT and ELISA seropositivity, along with prevalence ratios (PR), confidence intervals (CI), and p values.

3. Revised captions and narrative text to explicitly state that Tables 1 and 2 show marginal distributions, while Tables 4 and 5 show conditional distributions.

4. Specified the exact statistical tests used (Pearson’s chi square or Fisher’s exact test where expected cell counts were <5) in the captions of Tables 4 and 5, ensuring transparency and reproducibility.

We believe this structure maintains clarity, avoids redundancy, and ensures that both marginal and conditional distributions are clearly presented in the manuscript.

---

## [Editor Report · Decision Letter 3]

16 Mar 2026

Toxoplasma gondii seropositivity among patients with sickle cell disease: Prevalence and Association with Blood Transfusion History

PONE-D-25-53926R3

Dear Dr. orish,

We’re pleased to inform you that your manuscript has been judged scientifically suitable for publication and will be formally accepted for publication once it meets all outstanding technical requirements.

Kind regards,

Masoud Foroutan, PhD

Academic Editor

PLOS One
---

## [Editor Report · Acceptance letter]

PONE-D-25-53926R3

PLOS One

Dear Dr. orish,

I'm pleased to inform you that your manuscript has been deemed suitable for publication in PLOS One. Congratulations! Your manuscript is now being handed over to our production team.

Kind regards,

on behalf of

Dr Masoud Foroutan

Academic Editor

PLOS One